# The effectiveness of community engagement using M-Mama champions in improving awareness of obstetric danger signs, birth preparedness and complication readiness among pregnant women in Bahi, Dodoma: A cluster randomized pragmatic implementation trial

Alex Sanga[1]*, Stephen Kibusi[1], James Tumaini Kengia[1,2]

**1** The University of Dodoma, School of Nursing and Public Health, Dodoma, Tanzania, **2** President's Office Regional Administration and Local Government, Dodoma, Tanzania

* sanga.alex@gmail.com

## Abstract

### Background

Maternal mortality remains a catastrophic condition for reproductive-age women in Tanzania. Inadequate ANC visits impair the efficiency of education programs for pregnant women on obstetric danger signs, birth preparedness and complication readiness, hence negatively influencing health behaviour and decision-making processes, contributing to maternal mortality. In this case, a complementary health education intervention for pregnant women in the community is necessary. M-MAMA Champions were introduced to determine their effectiveness in creating awareness of obstetric danger signs, birth preparedness complication readiness and their practice to complement the health system.

### Methods

A parallel arms, cluster-randomized pragmatic implementation trial, whereby pregnant women were recruited from four clusters and randomised at a ratio of 1:1. M-MAMA Champions empowered pregnant women with obstetric danger signs, birth preparedness and complication readiness in addition to standard care. Difference-in-difference analysis determined the intervention's effect.

### Results

The majority of 124 pregnant women recruited from intervention (N = 60) and control (N = 64) arms, were aged 16 to 19, 31.3% (n = 20) vs 20 to 24, 35.0% (n = 21) and had primary education 48.4% (n = 31) vs 51.7% (n = 31) in the control and intervention arms respectively. Awareness of Obstetric danger signs, birth preparedness complication readiness

**Data availability statement:** The data that support the findings of this study will be made available from the University of Dodoma upon reasonable request and with permission from the University of Dodoma. Data can be requested from the Directorate of Postgraduate Studies via info@udom.ac.tz or (+255) 262 310 003.

**Funding:** This research was supported by the Ministry of Health, Tanzania (Ufadhili wa Masomo 2022/2023). The funders had no role in study design, data collection and analysis, decision to publish, or preparation of the manuscript.

**Competing interests:** The authors have declared that no competing interests exist.

and their practice improved significantly by 64.2%, (P < 0.0001), 26.3%, (P = 0.00190 and 33.9%, (P = 0.0006) respectively.

## Conclusion

M-MAMA Champions, the facilitators of women groups in community engagement are effective in improving awareness of obstetric danger signs, birth preparedness and complication readiness among pregnant women. It's, therefore worth an adoption for wider application.

## Trial registration

NCT06325319 (Effect of Community Engagement Using M-Mama Champions), registered on 15th March 2024.

## Background

Maternal mortality remains a global public health issue with a global average of 256 per 100,000 live births maternal mortality ratio, despite the global initiative to reduce it to meet the targets under Sustainable Development Goal 3, which envisages reducing it to less than 70 for every 100,000 live births by 2030 [1]. Tanzania's maternal mortality ratio is estimated to be 104 per 100,000 live births [2], which is still higher than the SDG target to be attained by 2030. Maternal mortality commonly occurs during intrapartum and the first day post-partum [3] and Obstetric danger signs account for more than 75% of all obstetric complications leading to maternal deaths [4,5]. However, there is still low knowledge of obstetric danger signs, birth preparedness, and complication readiness and their practice among pregnant women, which plays a significant role in the slow pace of maternal mortality reduction [6].

Two-thirds of global maternal deaths occurred in Sub-Saharan Africa in 2017[7]. Through a three-delay model, three factors contribute to maternal mortality, delay in deciding to seek care, access a health facility, and receive optimal care once one has arrived at a health facility for definitive care [8]. The low knowledge of obstetric danger signs among couples especially those with low literacy [9,10] is among the factors that lead to the first delay which ultimately leads to maternal mortality. It's essential to help the couples prepare in advance for childbirth and make prompt decisions in case of obstetric emergencies for such interventions targeting the first delay, work in tandem with interventions targeting the second and third delays [11–13].

Lack of knowledge of the importance of seeking medical attention during pregnancy and labour among pregnant women negatively influences health behaviour and decision-making processes [3]. Empowering women with knowledge of Obstetric Danger Signs (ODS) ensures that they take ownership of the decisions about their care at the right time without relying on others. Effective community-based interventions especially in rural settings [14,15], including the community health extension workers or health volunteers are vital to empower pregnant women to identify obstetric danger signs in pregnancy and make prompt decisions to seek care, to bridge the gap for those who don't attend the minimum ANC contacts [16].

Women groups, another effective approach to empowering pregnant women with health literacy, are facilitators of learning through participatory learning and action (PLA) and have been able to reduce maternal deaths by up to 88% [14]. M-MAMA Champions, the innovation designed to provide health education in women groups coupled with their experience from M-MAMA referral and emergency services was tested in this study, to ensure that

knowledge on obstetric danger signs among pregnant women is enhanced and contributes towards birth preparedness and complication readiness and maternal mortality reduction [12,17]. The community engagement using M-MAMA Champions innovation was tested in this study to determine if its effectiveness in improving awareness of obstetric danger signs (ODS), birth preparedness and complication readiness (BPCR) among pregnant women is superior to routine approaches alone.

## Aim of the study

The study aimed to determine the effectiveness of community engagement using M-MAMA Champions on Awareness of Obstetric Danger Signs, Birth Preparedness and Complication Readiness and its practice among Pregnant Women in Bahi, Dodoma.

The specific objectives;

i.   To determine the effectiveness of community engagement using M-MAMA Champions on awareness of Obstetric Danger Signs among pregnant women.

ii.  To determine the effectiveness of community engagement using M-MAMA Champions on awareness of birth preparedness and complication readiness among pregnant women.

iii. To determine the effectiveness of community engagement using M-MAMA Champions on the practice of birth preparedness and complication readiness among pregnant women.

## Methods

### Study design

The cluster randomised pragmatic implementation trial was used for this study. Four [4] matched wards were purposively selected based on these criteria; wards with at least two health facilities (1 Health Center and 1 Dispensary or 2 Dispensaries). Both health facilities with Post-natal mothers who benefited from the M-MAMA emergency and referral program and are located in a rural setting. The selected primary healthcare facilities in the rural setting ensured equitable distribution of their characteristics regarding their accessibility. Nearly two-thirds of pregnant women in all wards experience more than one-hour walking distance to the nearby healthcare facility. The distance to the nearest health facility determines the attendance likelihood of pregnant women for ANC and intervention delivery, this would ultimately influence the intervention outcome for it was delivered at a health facility [18] The facilities with records of post-natal mothers who had received M-MAMA referral and the emergency serviceas a potential human resource for intervention delivery were vital in ensuring standardized features of all the randomized clusters. Therefore, primary healthcare facilities with the highest number of legible post-natal mothers were included in this study. The inclusion of dispensaries and or health centres from both arms was necessary for baseline comparability, the study population representativeness and services' quality variability across facility levels would determine the ANC utility among pregnant women [19] and influence the intervention compliance. The clusters were randomized at a ratio of 1:1, whereby, the intervention arm received the intervention plus the standard care for one (1) month meanwhile the control arm received only the standard care. The selected wards were Mundemu which is located about 88km driving distance from Bahi Council headquarters, Lamaiti (28km), Mpalanga (90km) and Chipanga (63km). The wards are situated in locations where study participants utilize the facilities within their localities for routine services including ANC, unless for conditions requiring emergency care at a referral point. Therefore, the intervention's efficacy was unlikely to be interfered by services provided at other health facilities as described in Fig 1.

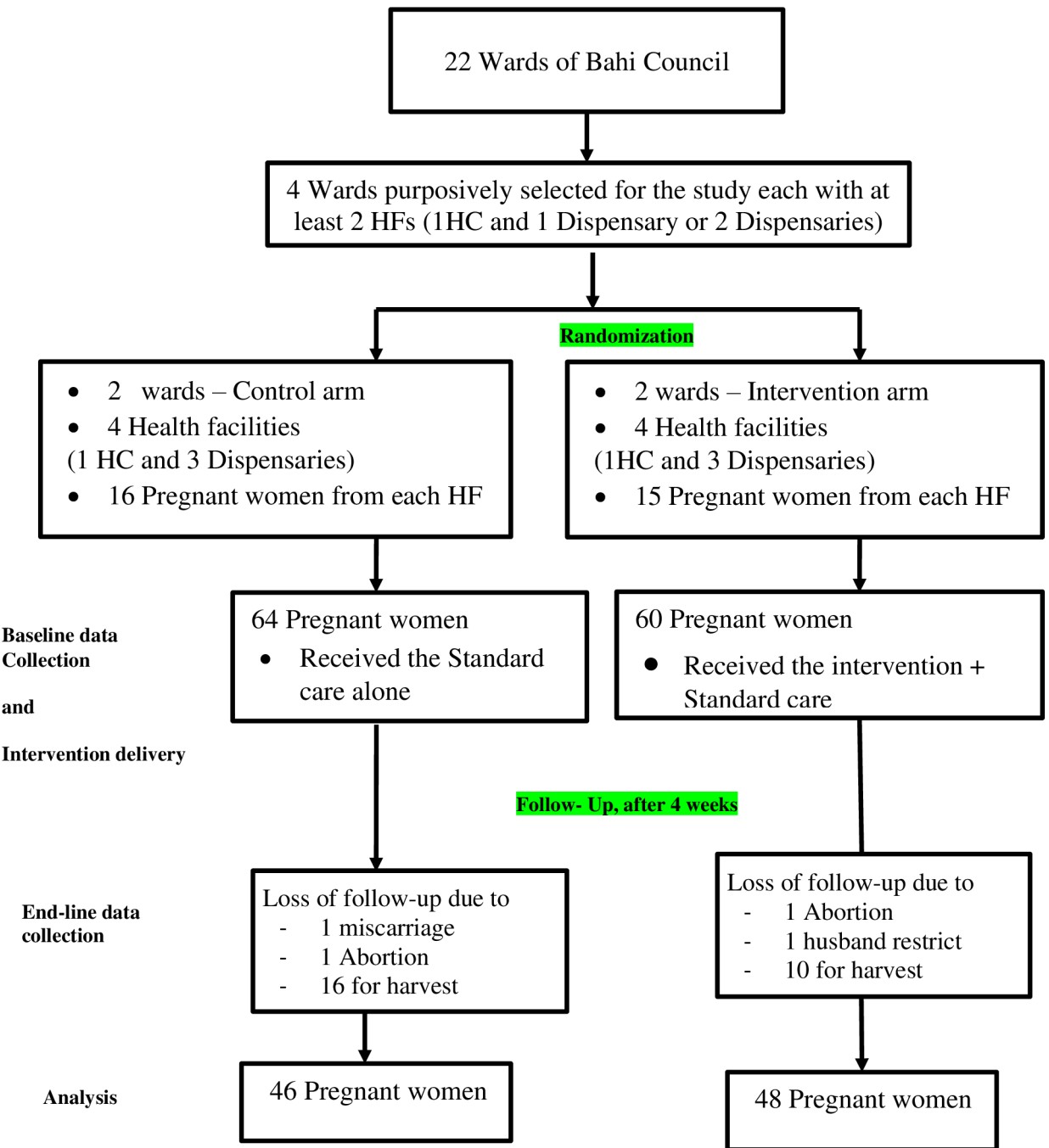

**Fig 1. The Consort Flow diagram of the cluster randomized pragmatic implementation trial, the clusters selection, randomization, baseline assessment, follow-up and end-line assessment.**

## Recruitment of participants

The study participants were recruited from the clusters' health facilities according to the inclusion criteria. The screening of eligible participants from the health facility registries of pregnant women who had booked for Antenatal Care was done before selection. Pregnant women with a gestational age below 28 weeks during screening were eligible participants, whereby 15

to 16 pregnant women were randomly selected through simple random sampling (using a random number generator on Excel) from each health facility.

**The study population** included all pregnant women available in the study area during the study period.

**Inclusion criteria**; pregnant women in the first and second trimesters (up to 28 weeks of GA). The two early trimesters included in the study were considered significant for early antenatal care contact initiation to create awareness of obstetric danger signs, birth preparedness and complication readiness to enhance its practice all through to delivery [20].

**Exclusion Criteria**; Pregnant women in the 1st and 2nd gestation age who were sick and admitted.

## Randomization

A block randomization was done at the ward level. A pre-determined number for each ward was at least 30 pregnant women (2 health facilities for each block) as per inclusion criteria. Four wards were randomised to either the intervention or control arm at a ratio of 1:1 through a computerized random number generator that was done using Excel, of which two (2) wards were randomised to the intervention and the other two (2) to the control arm.

## Intervention arm

The intervention arm received "The community engagement using M-MAMA Champions on Obstetric Danger Signs, Birth Preparedness, and Complication Readiness intervention" in addition to standard care. Two sessions, with an interval of four weeks, intervention was delivered using a customized and Kiswahili translated package, for one month from 15th April to 26th May 2024. Each session took at least 2 hours [21,22]. The utilized package was adapted from the Ministry of Health Tanzania package to empower Community Health Workers (CHWs).

## Control arm

The control arms didn't receive the intervention instead, they continued receiving the standard care. The standard care for pregnant women specifically in health education includes the package delivered by the healthcare workers at the Antenatal care clinics. The package should be delivered to the pregnant woman during every ANC visit.

## Recruitment and training of M-Mama champions

**Recruitment of M-MAMA champions.** The M-MAMA Champions were identified and recruited from the intervention-allocated clusters. The M-MAMA Champions' data were accessed from the health facility registries (referring or receiving health facilities) of post-natal mothers who had benefited from the M-MAMA program. The identified post-natal mothers were contacted through phone calls and requested to participate in the study. Those who were ready to participate in the study were visited by the CHW or village leader and were informed to visit the health facility on a specific day, for a collective study description and consent.

The M-MAMA Champions were required to have a minimum of the following criteria; 15 to 49 years of age, at least 3 months post-delivery, with at least primary education, ready and volunteering to empower other women. The socio-demographic data were collected from nine (9) recruited M-MAMA Champions before the collective orientation to the package used for empowering pregnant women with knowledge of obstetric danger signs, birth preparedness and complication readiness.

**Socio-demographic characteristics of M-MAMA Champions (N = 9).** The recruited M-MAMA Champions had a mean age of 24.78 ± 7.12 years (ranging from 18 to 37 years). Among them, 6 (66.7%) were peasants, 2 (22.2%) engaged in petty business and 1 (11.1%) was a housewife. All the M-MAMA Champions were religiously Christians and had permanent residence in the respective wards in the rural setting, whereby 6 (66.7%) were married and 3 (33.3%) were single, of whom 6 (66.7%) had attained secondary education and 3 (33.3%) had primary education. Most of the M-MAMA Champions had delivered once 4 (44.4%) or twice 4 (44.4%); only one had delivered more than 3 times. The M-MAMA Champions were socially grouped into those who were part of the women's association in the church or mosque 2 (22.2%), a member has shared to the Village Community Banks (VICOBA) 3 (33.3%) and the rest 4 (44.4%) did not belong to any of the women social group. Eight (88.9%) of the M-MAMA Champions were ready to deliver the intervention voluntarily but needed training before the intervention delivery except one who needed to be paid a small wage in addition.

**Orientation of M-Mama champions to the intervention.** The intervention package used by M-MAMA Champions for empowering pregnant women was first delivered by the principal investigator to M-MAMA Champions. The intervention package encompassed obstetric danger signs, birth preparedness and complication readiness. The intervention package was adopted from Tanzania's Community Health Workers (CHWs) package on reproductive and child health, which accommodated obstetric danger signs, birth preparedness and complication readiness.

This was a one-day orientation. It took place at a health facility of recruitment, whereby a specific room or space was secured for the orientation. The orientation was done whereby two sections of the package, ODS and BPCR were covered. It was done using the adapted package prepared in a brochure with pictures. 1 to 3 M-MAMA Champions per cluster were oriented on the intervention package and participated in the intervention for pregnant women.

**Sensitisation of pregnant women on ODS and BPCR.** The intervention on sensitisation of pregnant women on Obstetric Danger Signs, Birth Preparedness and Complication Readiness and their practice was delivered by M-MAMA Champions. The intervention was delivered in pregnant women groups of about 15 to 16 whereby one (1) M-AMA Champion acted as a facilitator per group using a friendly adapted intervention package. The package was designed into two sessions as indicated in Table 1, each session took two (2) hours. The sensitisation was done twice during the study period, with an interval of four weeks in between. The intervention was delivered through reading and discussions, questions and answers and demonstrations supported by pictures in the package (brochure) as summarized in Fig 2.

## Sample size estimation and sampling technique

**Sample size estimation.** The study assessed the effect of the M-MAMA champions in providing health education on obstetric danger signs whereby, the difference in the

Table 1. The content of the community obstetric sensitization package.

| Session | Content |
| --- | --- |
| First | The session covered Obstetric Danger Signs (ODS), which are likely to occur in pregnant women during pregnancy. It covered the introduction, meaning, list of danger signs and description, pictures of presentations for some danger signs and recommendations. |
| Second | The session covered Birth Preparedness and Complication Readiness (BPCR). It covered the introduction, a list of key items for birth preparedness, other items for preparedness fitting the local context (not included in the analysis) and recommendations. |

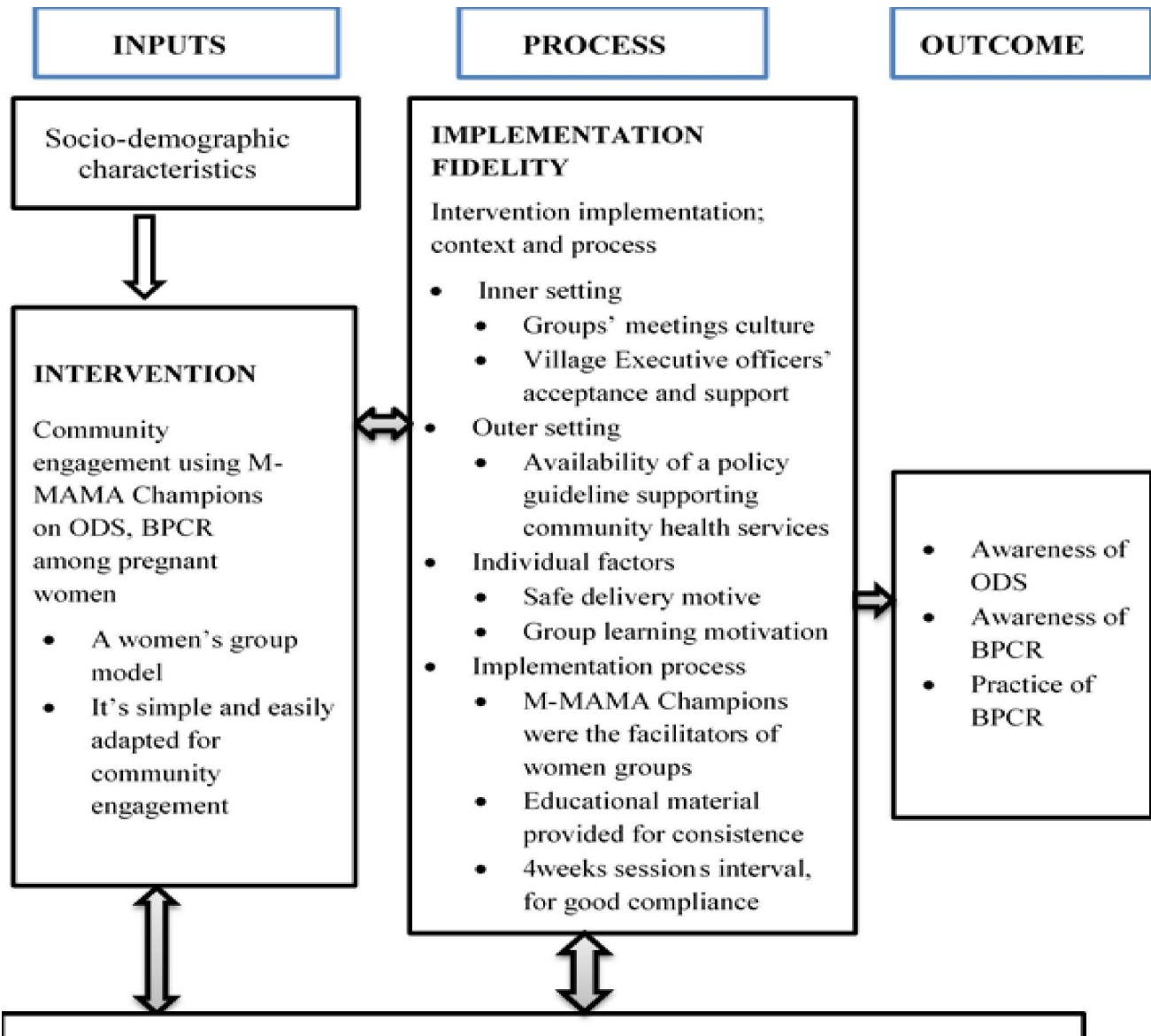

**Fig 2. Intervention implementation Conceptual Framework, with inputs (socio-demographics & intervention), processes for intervention delivery and outcomes expected from the intervention.**

proportions of knowledge among pregnant women was compared before and after the intervention and between arms. The sample size estimation was done to test the superiority hypothesis that the improvement in awareness will increase by at least 20% in the intervention

compared to the control arm. A formula for sample size to test the superiority hypothesis as recommended by [23] was used. Therefore, the sample size is highlighted below

$$N = \frac{(Z1-\alpha/2 + Z1-\beta)2 \ (P1(1-P1) + P2(1-P2)(1+(n-1)\rho))}{d^2}$$

Whereby

Z1-α/2 = Standard Normal Variate (for 5% type I error whereby p < 0.05, it's equal to 1.96)
Z1-β= Standard Normal variate for type II error (Power of 80% is equivalent to 0.84) [24]
P1 = The probability of an event in the control arm (0.3) [25]
P2 = The probability of an event in the treatment arm (set at 0.5)
d = The clinically important difference in intervention proportions (P1 - P2)
ρ = Intra-cluster correlation coefficient (0.015) [26]
n = Number of individuals per cluster (15)
N = Total number of individuals per arm

$$N = \frac{(1.96+0.84)2(0.3(1-0.3) + 0.5(1-0.5)(1+(15-1)0.015))}{(0.5-0.3)2}$$

$$N = \frac{7.84 \ x \ 0.21 + 0.25 \ x \ 1.21}{0.04}$$

$$N = \frac{1.65 + 0.3025}{0.04}$$

$$N = 49 \ \text{per arm}$$

The total sample size per arm plus a 19% attrition rate [27] is 60.

The study assumes a significance level of 5% and a power of 80%. The study subjects were randomly allocated at a ratio of 1:1.

**Sampling technique.** A multistage sampling technique was employed. The first stage was a purposive sampling of four (4) out of twenty-two (22) wards of Bahi District Council meeting these criteria; a ward with at least two health facilities (1HC and 1Dispensary or 2 Dispensaries) and located in the rural setting to ensure comparability between wards. The two health facilities from each ward were used for participants' recruitment. A simple random sampling of at least 15 eligible pregnant women from each health facility was done.

## Data collection method and instrument

**Data collection method.** An interview method using a structured questionnaire was used for data collection before and after the intervention. Data was collected from pregnant women by the principal investigator and the research assistants who were professionally holders of the Diploma in Clinical Medicine and were trained before data collection commenced. The recruitment health facility was used for Data collection, whereby a special room or space was secured for that exercise. This involved reading the questions or items from the data collection tool by the interviewer expecting to receive the responses from the participant depending on the nature of the question, whether it's a multiple choice, yes and no or an open-ended question. Baseline data was collected from the 8th to the 14th of April 2024, whereas end-line data was collected from the 16th to the 27th of May 2024.

**Data collection instrument.** A structured questionnaire adapted from the JHPIEGO birth preparedness and complication readiness monitoring and survey tool [28] was used for data collection. A few items specifically on Obstetric danger signs were added to the tool from other tools. The adapted tool was shared with various subject matter experts for comments before using it to ensure face and content validity.

The final adapted tool had four sections with sixteen (16), multiple choice, dichotomous and open-ended items on socio-demographic and obstetric characteristics, eight (8) dichotomous items on awareness of obstetric danger signs, five (5) dichotomous items on awareness of birth preparedness and complication readiness and five (5) dichotomous items on practice of birth preparedness and complication readiness, tested through a 10% the sample size and was determined to be reliable (Cronbach alpha 0.761). The tool was translated to Kiswahili, in collaboration between the subject matter experts and the supervisors for easy understanding by the research participants.

**Data analysis.** The collected data were entered, coded, cleaned and analysed using Statistical Package for Social Sciences (SPSS) version 27. The descriptive results were summarized using mean, standard deviation, range, proportions and tables. Inferential analysis was done through difference in difference (DID) using a Generalized Estimating Equation (GEE) to determine the intervention effect and the difference between the study arms over time. The confidence level was 95% and the significance level was 0.05 (p-value <0.05).

## Variables and variables measurement

**Variables definition.** Pregnant women's demographic characteristics and community engagement using M-MAMA Champions were the independent variables. The dependent (outcome) variables; awareness of obstetric danger signs, birth preparedness and complication readiness among Pregnant women were the primary outcomes and practice of birth preparedness and complication readiness was the secondary outcome.

The obstetric danger signs referred to in this manuscript include those which occur during pregnancy namely; - Abdominal pain, severe body fatigue, vaginal bleeding, fever, difficulty in breathing, persistent headache, blurred vision, swelling/oedema of hands, face or feet, foul smell discharge, unconsciousness, convulsion, reduced foetal movement and pallor.

Birth preparedness and complication readiness encompasses five (5) key components which include; 1) Identification of place to give birth, 2) Identification of a potential blood donor, 3) Identification and selection of a skilled birth attendant, 4) Identification and selection of means of transport in case of an emergency and 5) Saving money for emergency transportation [29].

**Variable measurement.** The socio-demographic characteristics were measured by three items (age, parity and ANC visits) on a numerical scale, two items (marital status and Obstetric complication history) on a nominal scale and one item (educational level) on an ordinal scale.

Awareness of obstetric danger signs was measured by eight (8) items on a binary scale. One point was awarded for the correct answer and zero for the wrong answer. A total score was eight (8), whereby a sum of correctly mentioned ODS was computed, those who scored three (3) or more were considered aware and have good knowledge whereas those who scored below three (3) were considered unaware [30].

Awareness of birth preparedness and complication readiness (BPCR) was measured by five (5) items on a binary scale. One point was awarded for the correct answer and zero for the wrong answer. A total score was five (5), those who were able to mention/identify at least three (3) out of five (5) BPCR components were regarded as being aware of BPCR [31]s.

Birth Preparedness and Complication Readiness practice was measured by five (5) items on a binary scale. One point was awarded for the accomplished practice and zero for non-accomplished practice. A total score was five (5), those who accomplished at least three (3) out of five (5) birth preparedness and complication readiness factors were regarded as having good practice of birth preparedness and complication readiness [31].

**Intervention fidelity.**  The study participants were recruited by the research assistants without disclosure of the treatment allocation. Randomization was done at the ward level to control for contamination though the number of clusters decreased. The observation of the intervention delivery was done by the principal investigator, research assistants and CHWs. M-MAMA Champions were oriented before the intervention delivery, the intervention package adopted from the Ministry of Health Tanzania, is standard and valid. The statistician who assisted with data analysis was blinded to the arms allocation. Those who were Lost during the end-line data collection, follow-upwas done through both phone calls by the researcher and research assistants and physical visits by the CHWs. A few had obstetric complications but, the majority prioritised harvest activity instead, especially among controls.

**Ethical consideration.**  The ethical clearance was granted by the Research Ethical Committee of the University of Dodoma (Ref. No. MA.84/261/02/'A'/69/115). Permission to conduct the study was provided by the President's Office, Regional Administration and Local Government (PO RALG) and all other authorities from regional administration, council administration, ward administration and the village and or health facility administration levels before data collection. Also, informed consent was sought from each participant before data collection commenced. Detailed information on the study aims and procedures, benefits and risks to participants by participating in the study and the participant's role in the study were provided to the participants. The study included participants who were aged below 18 years, however, all of them were already married, therefore, the ascent was provided by their husbands after the study description was provided. Some of the participants were unable to read and write, they consented by putting a thumb sign against their written name after reading aloud the informed consent witnessed by the healthcare worker from the healthcare facility where the participants were recruited. All other participants, provided informed consent before data collection commenced. The ethical principles of the Declaration of Helsinki by the World Medical Association were adhered to during data collection to ensure the protection of the participants' values, dignity and integrity.

## Results

### Socio-demographic characteristics of pregnant women

A total of 124 pregnant women were recruited in this study, whereby their randomization was done at a ratio of 1:1 in the intervention (60) and the control (64) arms. The age group of 16 to 19, 20 (31.3%) in the control arm and 20 to 24, 21 (35.0%) in the intervention arm constituted the majority in this study. The majority of pregnant women had attained primary education in both the control 31 (48.4%) and the intervention 31 (51.7%) arms. Surprisingly, this was followed by a significant proportion of those who had never gone to school 26 (40.6%) and 23 (38.3%) in the control and intervention arms respectively, whereas the rest had higher education levels. 52 (81.3%) pregnant women in the control arm and 54 (90.0%) in the intervention arm were married or cohabiting and the rest were single. The control arm had a higher proportion of pregnant women who had a parity of 3 and above, 23 (35.9%) followed by primi-gravida, 22 (34.4%) compared to the intervention arm which constituted the majority with a

parity of 1, 18 (30.0%) followed by primigravida, 16 (26.7%). The history of birth complications from previous pregnancies was reported by 18 (30.0%) and 14 (23.3%) among pregnant women in the control and intervention arms respectively. At the time of data collection, from 1st and 2nd trimester pregnant women, those who had made at least 4 - 5 Antenatal care visits were 5 (7.8%) in the control and 11 (18.3%) in the intervention arms. The socio-demographic characteristics have been summarized in Table 2.

## Awareness of obstetric danger signs

The itemized awareness on obstetric danger signs among pregnant women has been summarized in Table 3, whereby vaginal bleeding is a danger sign reported by the majority of pregnant women 42 (33.9%) and 62 (66.0%), persistent headache 31 (25.0%) and 42 (44.7%) and severe body fatigue 28 (22.6%) and 36 (38.3%) at both baseline and end-line respectively. Seizure 5 (4.0%), blurred vision 3 (2.4%), and loss of consciousness 1 (0.8%) were most likely to be reported by the pregnant women. However, the baseline awareness of persistent headache was 9 (14.1%) among controls compared to 22 (36.7%) in the intervention arm (P = 0.004). Also, awareness of Swelling of face, hands or feet was 8 (12.5%) among controls compared to 1 (1.7%) in the intervention arm (P = 0.02). This shows a significant difference in awareness of these particular items between arms as indicated in Fig 3.

**Table 2. Socio-demographic characteristics of the participants by treatment arm.**

| | Baseline | | | End-line | | |
|---|---|---|---|---|---|---|
| **Characteristics** | **Control** | **Intervention** | **P-Value** | **Control** | **Intervention** | **P-Value** |
| All | 64(51.6%) | 60(48.4%) | | 46(48.9%) | 48(51.1%) | |
| **Age** | | | 0.43 | | | 0.66 |
| 16-19 | 20 (31.3%) | 14 (23.3%) | | 16 (34.8%) | 12 (25.0%) | |
| 20-24 | 18 (28.1%) | 21 (35.0%) | | 14 (30.4%) | 14 (29.2%) | |
| 25-29 | 10 (15.6%) | 14 (23.3%) | | 9 (19.6%) | 11 (22.9%) | |
| 30+ | 16 (25.0%) | 11 (18.3%) | | 7 (15.2%) | 11 (22.9%) | |
| **Education level** | | | 0.94 | | | 0.45 |
| Never gone to school | 26 (40.6%) | 23 (38.3%) | | 18 (39.1%) | 19 (39.6%) | |
| Primary school | 31 (48.4%) | 31 (51.7%) | | 23 (50.0%) | 27 (56.3%) | |
| Secondary school | 7 (10.9%) | 6 (10.0%) | | 5 (10.9%) | 2 (4.2%) | |
| **Current Marital Status** | | | 0.17 | | | 0.053 |
| Not married | 12 (18.8%) | 6 (10.0%) | | 9 (19.6%) | 3 (6.3%) | |
| Married/Cohabiting | 52 (81.3%) | 54 (90.0%) | | 37 (80.4%) | 45 (93.8%) | |
| **Parity** | | | 0.037 | | | 0.58 |
| 0 | 22 (34.4%) | 16 (26.7%) | | 16 (34.8%) | 14 (29.2%) | |
| 1 | 8 (12.5%) | 18 (30.0%) | | 5 (10.9%) | 10 (20.8%) | |
| 2 | 11 (17.2%) | 14 (23.3%) | | 11 (23.9%) | 12 (25.0%) | |
| >3 | 23 (35.9%) | 12 (20.0%) | | 14 (30.4%) | 12 (25.0%) | |
| **Experienced birth complications** | | | 0.68 | | | 0.71 |
| No | 51 (79.7%) | 46 (76.7%) | | 36 (78.3%) | 36 (75.0%) | |
| Yes | 13 (20.3%) | 14 (23.3%) | | 10 (21.7%) | 12 (25.0%) | |
| **Number of ANC Visits** | | | 0.081 | | | 0.42 |
| 1-3 | 59 (92.2%) | 49 (81.7%) | | 41 (89.1%) | 40 (83.3%) | |
| 4-5 | 5 (7.8%) | 11 (18.3%) | | 5 (10.9%) | 8 (16.7%) | |

**Table 3. The relationship between the intervention and the reported awareness about obstetric Danger Signs among pregnant women (Intervention N = 48, Control N = 46).**

| Items | Baseline | | | | End line | | | |
|---|---|---|---|---|---|---|---|---|
| | Total | Control | Intervention | P-Value | Total | Control | Intervention | P-Value |
| | N = 124 | N = 64 | N = 60 | | N = 94 | N = 46 | N = 48 | |
| Severe body fatigue | | | | 0.81 | | | | 0.001 |
| Yes | 28 (22.6%) | 15 (23.4%) | 13 (21.7%) | | 36 (38.3%) | 10 (21.7%) | 26 (54.2%) | |
| No | 96 (77.4%) | 49 (76.6%) | 47 (78.3%) | | 58 (61.7%) | 36 (78.3%) | 22 (45.8%) | |
| Vaginal Bleeding | | | | 0.31 | | | | 0.006 |
| Yes | 42 (33.9%) | 19 (29.7%) | 23 (38.3%) | | 62 (66.0%) | 24 (52.2%) | 38 (79.2%) | |
| No | 82 (66.1%) | 45 (70.3%) | 37 (61.7%) | | 32 (34.0%) | 22 (47.8%) | 10 (20.8%) | |
| Presence of Fever | | | | 0.69 | | | | 0.10 |
| Yes | 19 (15.3%) | 9 (14.1%) | 10 (16.7%) | | 21 (22.3%) | 7 (15.2%) | 14 (29.2%) | |
| No | 105 (84.7%) | 55 (85.9%) | 50 (83.3%) | | 73 (77.7%) | 39 (84.8%) | 34 (70.8%) | |
| Persistent headache | | | | 0.004 | | | | <0.001 |
| Yes | 31 (25.0%) | 9 (14.1%) | 22 (36.7%) | | 42 (44.7%) | 11 (23.9%) | 31 (64.6%) | |
| No | 93 (75.0%) | 55 (85.9%) | 38 (63.3%) | | 52 (55.3%) | 35 (76.1%) | 17 (35.4%) | |
| Blurred vision | | | | 0.60 | | | | 0.009 |
| Yes | 3 (2.4%) | 2 (3.1%) | 1 (1.7%) | | 10 (10.6%) | 1 (2.2%) | 9 (18.8%) | |
| No | 121 (97.6%) | 62 (96.9%) | 59 (98.3%) | | 84 (89.4%) | 45 (97.8%) | 39 (81.3%) | |
| Swelling of face, hands or feet | | | | 0.020 | | | | <0.001 |
| Yes | 9 (7.3%) | 8 (12.5%) | 1 (1.7%) | | 34 (36.2%) | 5 (10.9%) | 29 (60.4%) | |
| No | 115 (92.7%) | 56 (87.5%) | 59 (98.3%) | | 60 (63.8%) | 41 (89.1%) | 19 (39.6%) | |
| Loss of consciousness | | | | 0.30 | | | | <0.001 |
| Yes | 1 (0.8%) | 0 (0.0%) | 1 (1.7%) | | 13 (13.8%) | 0 (0.0%) | 13 (27.1%) | |
| No | 122 (99.2%) | 63 (100.0%) | 59 (98.3%) | | 81 (86.2%) | 46 (100.0%) | 35 (72.9%) | |
| Seizure | | | | 0.60 | | | | <0.001 |
| Yes | 5 (4.0%) | 2 (3.1%) | 3 (5.0%) | | 20 (21.3%) | 0 (0.0%) | 20 (41.7%) | |
| No | 119 (96.0%) | 62 (96.9%) | 57 (95.0%) | | 74 (78.7%) | 46 (100.0%) | 28 (58.3%) | |

## Generalized estimating equation analysis about awareness of obstetric danger signs

The estimate for the intervention arm is 0.2815 with a p-value of 0.5824 (not significant) indicating that, there is no significant difference in awareness of obstetric danger signs between the intervention and control arms at baseline. Similarly, the results showed that there is no significant change in awareness from baseline to end-line among subjects in the control arm ($\beta$ = 0.2411, $p$ = 0.633). Besides, the interaction term (Treatment*Time) has an estimate of 3.3878 with a p-value of <0.0001, indicating a positive significant interaction effect. This suggests that the intervention arm showed a significant increase in awareness of obstetric danger signs from baseline to End-line as compared to the control arm with a Cohen's D (effect size) of 1.77 (large effect) as indicated in Table 4. Therefore, community engagement intervention using M-MAMA Champions was significantly effective in improving the awareness of ODS among pregnant women, from 15% to 79.2%.

## Awareness about birth preparedness and complication readiness

Awareness of securing emergency money for delivery 33 (26.6%) and identification of emergency transport 25 (20.2%) were the most frequently identified items out of five items

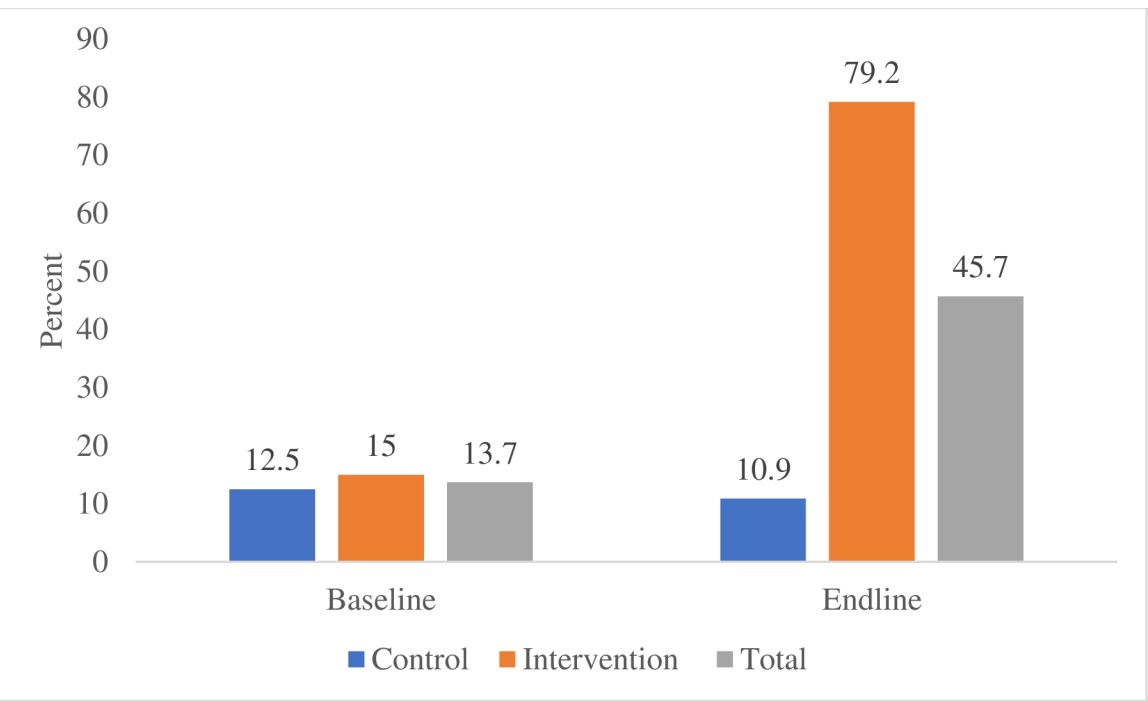

**Fig 3. The proportion of pregnant women aware of at least three out of eight obstetric danger signs at baseline and End-line among pregnant women who participated in the trial by treatment arm.**

**Table 4. The parameter Estimates of the GEE model for the effectiveness of community engagement using M-MAMA Champions about awareness of Obstetric Danger Signs.**

| Parameter | Estimate ($\beta$) | Standard Error | 95% Confidence Interval | | P-Value | Cohen's d |
|---|---|---|---|---|---|---|
| Intercept | −2.5276 | 0.6077 | −3.7187 | −1.3364 | <0.0001 | |
| **Treatment** | | | | | | |
| Intervention | 0.2815 | 0.5121 | −0.7221 | 1.2851 | 0.5824 | |
| Control | Reference | | | | | |
| **Time** | | | | | | |
| End-line | −0.2411 | 0.5049 | −1.2307 | 0.7484 | 0.633 | |
| Baseline | Reference | | | | | |
| **Treatment\*-Time** | 3.3878 | 0.6957 | 2.0243 | 4.7513 | <0.0001 | 1.77 |
| Parity | 0.2168 | 0.1979 | −0.1711 | 0.6047 | 0.2733 | |

as crucial to birth preparedness and complication readiness among pregnant women in Bahi, Dodoma. However, identification of a compatible blood donor 0(0%) and a trained birth attendant to assist delivery 0(0%) were the reported items among pregnant women in both intervention and control arms before and after the intervention as indicated in Table 5.

From the GEE model, the results suggest that, at baseline, there is no significant difference in the awareness of birth preparedness and complication readiness between the intervention and control arms (β = 0.1865, p = 0.7519). The estimate of the time effect is 0.0252 with a p-value of 0.0009 (significant), indicating a significant increase in awareness from baseline to end line among subjects in the intervention arm.

**Table 5. The relationship between the intervention and the reported awareness about birth preparedness and complication readiness among pregnant women (Intervention N = 48, Control N = 46).**

| Characteristic | Baseline | | | | End-line | | | |
|---|---|---|---|---|---|---|---|---|
| | Total | Control | Intervention | P-Value | Total | Control | Intervention | P-Value |
| | N = 124 | N = 64 | N = 60 | | N = 94 | N = 46 | N = 48 | |
| Health Facility where delivery will take place | | | | 0.52 | | | | 0.37 |
| Yes | 20 (16.1%) | 9 (14.1%) | 11 (18.3%) | | 20 (21.3%) | 8 (17.4%) | 12 (25.0%) | |
| No | 104 (83.9%) | 55 (85.9%) | 49 (81.7%) | | 74 (78.7%) | 38 (82.6%) | 36 (75.0%) | |
| Emergency transport to the health facility where delivery will take place | | | | 0.19 | | | | <0.001 |
| Yes | 25 (20.2%) | 10 (15.6%) | 15 (25.0%) | | 42 (44.7%) | 8 (17.4%) | 34 (70.8%) | |
| No | 99 (79.8%) | 54 (84.4%) | 45 (75.0%) | | 52 (55.3%) | 38 (82.6%) | 14 (29.2%) | |
| Emergency money for delivery | | | | 0.10 | | | | <0.001 |
| Yes | 33 (26.6%) | 13 (20.3%) | 20 (33.3%) | | 56 (59.6%) | 14 (30.4%) | 42 (87.5%) | |
| No | 91 (73.4%) | 51 (79.7%) | 40 (66.7%) | | 38 (40.4%) | 32 (69.6%) | 6 (12.5%) | |
| Blood donor for delivery-related transfusion | | | | | | | | <0.001 |
| Yes | 0 (0%) | 0 (%) | 0 (0%) | | 14 (14.9%) | 1 (2.2%) | 13 (27.1%) | |
| No | 124 (100.0%) | 64 (100.0%) | 60 (100.0%) | | 80 (85.1%) | 45 (97.8%) | 35 (72.9%) | |
| Trained birth attendant to assist with delivery | | | | | | | | 0.013 |
| Yes | 0 (0%) | 0 (%) | 0 (0%) | | 6 (6.4%) | 0 (0.0%) | 6 (12.5%) | |
| No | 124 (100.0%) | 64 (100.0%) | 60 (100.0%) | | 88 (93.6%) | 46 (100.0%) | 42 (87.5%) | |

The interaction term has an estimate of 1.2545 with a p-value of 0.0019, indicating a significant positive interaction effect. This indicates that the intervention arm showed a significant increase in awareness of birth preparedness and complication readiness at the end line compared to the control arm with a Cohen's D (effect size) of 1.49 (large effect size). Thus, the community engagement intervention using M-MAMA Champions is effective in increasing awareness of birth preparedness and complication readiness among pregnant women from 15.2% to 39.6% in the intervention arm as highlighted in Fig 4.

## Practice of birth preparedness and complication readiness

The results indicate that securing emergency money, 28 (22.6%) versus 47 (50.0%), identified an emergency transport 20 (16.1%) versus 30 (31.9%) and identified health facility where the delivery would take place 18 (14.5%) versus 26 (27.7%) are the items mostly reported by participants in this study before and after the intervention respectively. Identification of a blood donor 2 (1.6%) and trained birth attendant 0 (0%), were the list reported items as summarized in Table 6.

The GEE model results as shown in Table 7, the estimate of the intervention arm is 0.1865 with a p-value of 0.7519 (not significant), suggesting that there is no significant difference in the baseline practice of birth preparedness and complication readiness between the intervention and control arms. This indicates that, at baseline, the proportion of women noted to be well prepared on birth preparedness and complication readiness was comparable for intervention and control arms.

The estimate of the end-line is 0.1316 with a p-value of 0.5213 (not significant), indicating no significant change in the practice of birth preparedness and complication readiness from baseline to End-line among subjects in the control arm.

Moreover, the interaction term has an estimate of 1.4519 with a p-value of 0.0006, indicating a significant positive interaction effect. This suggests that the intervention arm showed a

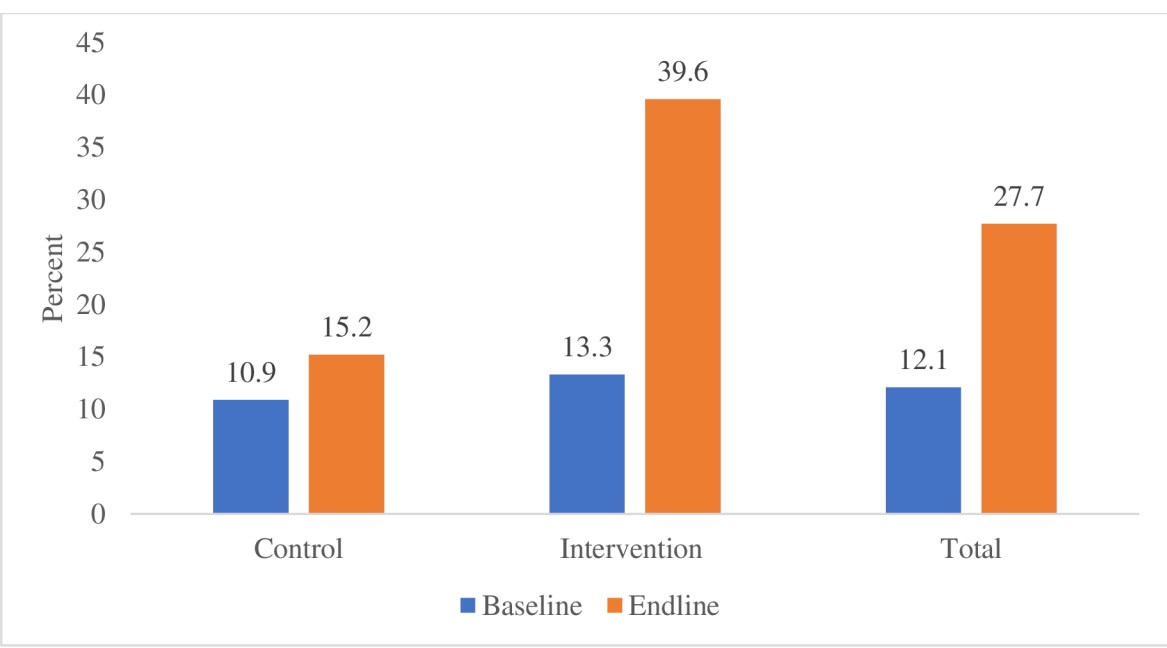

**Fig 4. The proportion of pregnant women aware of at least three out of five items for birth preparedness and complication readiness among pregnant women who participated in the trial by treatment arms.**

**Table 6. The relationship between the intervention and the reported practice of birth preparedness and complication readiness among pregnant women (Intervention N = 48, Control N = 46).**

| | Baseline | | | | End-line | | | |
|---|---|---|---|---|---|---|---|---|
| | Total | Control | Intervention | P-Value | Total | Control | Intervention | P-Value |
| | N = 124 | N = 64 | N = 60 | | N = 94 | N = 46 | N = 48 | |
| Health Facility identified | | | | 0.51 | | | | 0.008 |
| Yes | 18 (14.5%) | 8 (12.5%) | 10 (16.7%) | | 26 (27.7%) | 7 (15.2%) | 19 (39.6%) | |
| No | 106 (85.5%) | 56 (87.5%) | 50 (83.3%) | | 68 (72.3%) | 39 (84.8%) | 29 (60.4%) | |
| Emergency transport to the Health Facility secured | | | | 0.87 | | | | <0.001 |
| Yes | 20 (16.1%) | 10 (15.6%) | 10 (16.7%) | | 30 (31.9%) | 7 (15.2%) | 23 (47.9%) | |
| No | 104 (83.9%) | 54 (84.4%) | 50 (83.3%) | | 64 (68.1%) | 39 (84.8%) | 25 (52.1%) | |
| Emergency money for delivery secured | | | | 0.53 | | | | <0.001 |
| Yes | 28 (22.6%) | 13 (20.3%) | 15 (25.0%) | | 47 (50.0%) | 9 (19.6%) | 38 (79.2%) | |
| No | 96 (77.4%) | 51 (79.7%) | 45 (75.0%) | | 47 (50.0%) | 37 (80.4%) | 10 (20.8%) | |
| Blood donor for delivery-related transfusion identified | | | | 0.96 | | | | 0.085 |
| Yes | 2 (1.6%) | 1 (1.6%) | 1 (1.7%) | | 3 (3.2%) | 0 (0.0%) | 3 (6.3%) | |
| No | 122 (98.4%) | 63 (98.4%) | 59 (98.3%) | | 91 (96.8%) | 46 (100.0%) | 45 (93.8%) | |
| Trained HCW to assist delivery identified | | | | | | | | |
| Yes | 0(0%) | 0(%) | 0(0%) | | 0(0%) | 0(%) | 0(0%) | |
| No | 124 (100.0%) | 64 (100.0%) | 60 (100.0%) | | 94 (100.0%) | 46 (100.0%) | 48 (100.0%) | |

significant improvement from 15% to 43.8% in the practice of BPCR from baseline to end-line compared to the control arm with an effect size of 1.28 (large effect size) as highlighted in Fig 5. Socio-demographic (age, education level, marital status, parity, ANC visits) and socio-economic (type of living house, drinking water source, livestock ownership, radio ownership,

**Table 7. The parameter Estimates of the GEE model for the effectiveness of community engagement using M-MAMA Champions on awareness of BPCR.**

| Parameter | Estimate | Standard Error | 95% Confidence Interval | | P-Value | Cohen's d |
|---|---|---|---|---|---|---|
| Intercept | −3.0536 | 1.1423 | −5.2924 | −0.8148 | 0.0075 | |
| **Treatment** | | | | | | |
| Intervention | 0.1865 | 0.59 | -0.9699 | 1.3429 | 0.7519 | |
| Control | Reference | | | | | |
| **Time** | | | | | | |
| End-line | 0.0252 | 0.0076 | 0.0103 | 0.0401 | 0.0009 | |
| Baseline | Reference | | | | | |
| **Treatment*Time** | 1.2545 | 0.403 | 0.4647 | 2.0443 | 0.0019 | 1.49 |
| Parity | 0.3459 | 0.3494 | −0.3388 | 1.0306 | 0.3222 | |

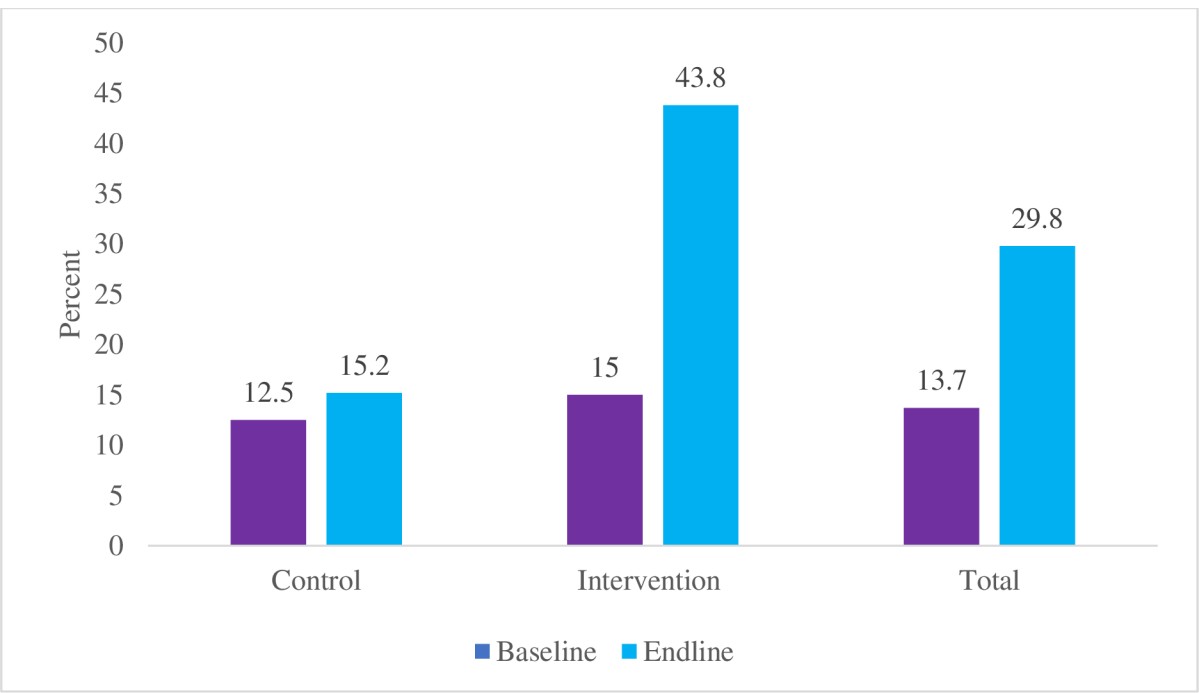

**Fig 5. The proportion of pregnant women who prepared at least three out of five items of birth preparedness and complication readiness among those who participated in the trial by treatment arms.**

mobile phone ownership, means of transport commonly used and distance to the nearest health facility) variables determined to be potential confounders were analysed, however, none were determined to a significant predictor of the outcomes of interest in this study.

## Discussion

This study aimed to determine the effectiveness of community engagement using M-MAMA Champions in improving the awareness of obstetric danger signs, birth preparedness and complication readiness among pregnant women, it therefore tested the superiority hypothesis of the intervention compared to standard care alone. Therefore, the determination of awareness of ODS, BPCR and their practice was the main aim of this study.

The study results confirm the hypothesis that community engagement using M-MAMA Champions in improving awareness of obstetric danger signs, birth preparedness and complication readiness among pregnant women in Bahi, Dodoma is more effective than the standard care alone. In this study, awareness of obstetric danger signs, birth preparedness complication readiness and practice of BPCR increased significantly in the intervention arm compared to comparisons after the intervention

## The awareness of obstetric danger signs among pregnant women

The awareness of obstetric danger signs among pregnant women is catalytic to enhance birth preparedness and complication readiness practice [32]. In this study, awareness of obstetric danger signs significantly increased in the intervention arm as compared to the comparisons after the intervention. Awareness of ODS in the intervention arm had a higher preponderance of response to vaginal bleeding, followed by persistent headache and severe body fatigue. Vaginal bleeding was the most commonly identified ODS among pregnant women, as reported in other studies. Nevertheless, blurred vision and loss of consciousness were least likely to be reported by the majority of pregnant women [33].

However, at baseline, the overall awareness of obstetric danger signs among pregnant women in Bahi Council was 13.7%. This means that only 13.7% of pregnant women were able to name at least three (3) obstetric danger signs during pregnancy without any additional intervention to standard care. This finding is similar to 14.8%, reported in a survey on Birth preparedness and complication readiness among women who became pregnant and or gave birth two years preceding the survey in Mpwapwa district, Tanzania [34]. The similarity of the two studies might have been attributed to the similarity of the study context, whereby the two councils are only separated by Chamwino Council in between.

Moreover, the observed finding is higher than the 6.6% reported in a cross-sectional study on Birth preparedness and complication readiness among pregnant women admitted to a rural hospital in Rwanda [35]. The difference might have been attributed to the difference in of the study participants. Also, the routine interventions in place for awareness of ODS between the two East African countries might be slightly different.

Also, a cross-sectional study on knowledge of obstetric danger signs among recently delivered women in Chamwino district, Tanzania [36] and a facility-based cross-sectional study on knowledge about birth preparedness and complication readiness and associated factors among primigravida women in Addis Ababa governmental health facilities (32) reported a higher awareness of 25.2% and 26.8% respectively. The low level of awareness of ODS among pregnant women in Bahi, might have been attributed to the low level of education (nearly 40% had never gone to school) and few ANC visits during pregnancy (<18.3% had attended 4-5 ANC visits), although the two variables were not significant predictors of awareness in this study. The overall awareness of ODS among pregnant women in Bahi, is very low.

Community engagement using M-MAMA Champions, in improving awareness of obstetric danger signs significantly improved awareness of ODS among pregnant women in the intervention compared to the control arm after intervention. The intervention enabled pregnant women to discuss the key obstetric danger signs, how to identify them, and actions to be taken once identified with the aid of a brochure that was provided to every pregnant woman. Further clarification of the package offered by the M-MAMA Champions as facilitators rendered a clear understanding that also helped the pregnant women unable to read and write. Awareness of obstetric danger signs among pregnant women had a significant increase by 64.2% from 15% to 79.2%, whereby, pregnant women in the intervention arm were more

likely to name at least three obstetric danger signs during pregnancy compared to those in the control arm after the intervention.

The findings imply that community engagement using M-MAMA Champions is more effective than the standard care alone in improving awareness of obstetric danger signs, among pregnant women, therefore it's worth an adoption for larger scope coverage

## The awareness of birth preparedness and complication readiness among pregnant women

Birth preparedness and complication readiness awareness among pregnant women is crucial for a tailored practice for improved maternal health through increased skilled healthcare utilization [37]. In this study, the awareness of birth preparedness and complication readiness among pregnant women in the intervention arm significantly increased after the intervention.

The response to BPCR items had a higher preponderance of awareness of securing emergency money for delivery and identification of emergency transport as key items for birth preparedness and complication readiness. However, the identification of a compatible blood donor and a trained birth attendant to assist delivery were the least likely items to be reported by pregnant women even after the intervention. Even though skilled birth attendance in Tanzania is estimated to be 85% (2), still pregnant women couldn't identify an SBA to assist delivery as vital to birth preparedness and complication readiness. This might be attributed to the shortage of human resources for health, which doesn't guarantee a pregnant woman to select a skilled birth attendant to assist with delivery,

At baseline overall awareness of birth preparedness and complication readiness was 12.1%. This finding is similar to the 13.7% reported in Babati on a community-based cross-sectional study of Does knowledge of danger signs influence the use of maternal health services among rural women? Findings from Babati Rural district, Northern Tanzania [38]s or 14% reported in Gambia on Pregnant women's awareness of antenatal danger signs and birth preparedness in rural Gambia as cited in [39]. The similarity might have been attributed to the context of which both were conducted in rural settings which are likely to be similar.

This finding is lower than the 22.7% that was reported in a cross-sectional descriptive community-based study on knowledge and practice of birth preparedness and complication readiness among pregnant women in the selected ward of Biratnager Municipality, NEPAL [40]s. However, the finding is higher than the 7.3% reported in a cross-sectional study on knowledge of birth preparedness and complication readiness among expecting couples in rural Tanzania, differences by sex [41] The difference might have been attributed to the methodological differences employed in the study and the geographical differences.

The low awareness of BPCR among pregnant women in Bahi, could be attributed to the inadequate or no health education sessions delivered by the healthcare providers or the community health workers to pregnant women on BPCR due to a huge human resource for health shortage and high client burden [42]. Also, the inadequate ANC attendance might have contributed to low BPCR awareness coupled with the illiterate majority of pregnant women.

The community engagement using M-MAMA champions on awareness of birth preparedness and complication readiness among pregnant women, in Bahi Dodoma, was determined to be effective in improving their awareness of BPCR. The intervention delivered significantly influenced the increase of awareness in the intervention arm different from the control arm. This concurs with the meta-analysis [43] on the effectiveness of women groups in enhancing health literacy among pregnant women. This shows the contribution of the intervention delivery model using M-MAMA Champions as facilitators to instil awareness among pregnant women on BPCR.

## The reported practice of birth preparedness and complication readiness among pregnant women

Birth preparedness and complication readiness, despite being considered a simple and practicable means of reducing maternal mortality, it is not widely practised by pregnant women evidenced by the slow decline of maternal mortality ratio [44]. It's estimated that more than three-quarters of maternal deaths would have been averted if all pregnant women could access a skilled birth attendant during delivery [20], the target yet to be reached by most countries including Tanzania.

During the study, the reported practice of BPCR significantly increased among pregnant women in the intervention arm after the intervention. Secured funds for delivery services and identified means of transport in case of emergency were the two items that most of the pregnant women had already prepared for after the intervention similar to the findings by [45] in Tanzania and Kenya. However, identification of the compatible blood donor and selection of the Skilled Birth Attendant (SBA) were the items the majority of pregnant women had not been prepared for even after the intervention.

The poor blood donor selection practice correlates to the 4.6% reported in a community-based, cross-sectional study on Birth preparedness, complication readiness, and associated factors among pregnant women in South Wollo Zone, Northeast Ethiopia [46]. The low practice of the identification of a skilled birth attendant might be attributed to the great shortage of human resources for health (66%) [42] in the country which doesn't guarantee that pregnant women to identify a trained birth attendant upon delivery hence, whoever is available is apt to provide the service.

The study's findings showed that, at baseline, the overall practice of BPCR was only 13.7% among all pregnant women. The finding is lower than the 18.8% that was reported in a facility-based cross-sectional study on awareness and practice of birth preparedness and complication readiness among pregnant women in the Bamenda Health District, Cameroon [44] The difference might have been attributed to the cultural differences between the two geographical areas as well as varying interventions to enhance the practice of BPCR.

However, this was higher than the 9.7% reported in a community-based cross-sectional survey on Birth preparedness and complication readiness among women of reproductive age in Kenya and Tanzania [45] and 9.3% that was reported in a cross-sectional descriptive community-based study on knowledge and practice of birth preparedness and complication readiness among pregnant women in the selected ward of Biratnager Municipality, NEPAL [40]. The differences might have been attributed to the availability and good utilization of community-based services in addition to facility-based services like Community Health workers who might have contributed to higher practice in Bahi, Dodoma, despite that it's still low.

For a pregnant woman to be considered well prepared, was supposed to have prepared at least three items key for birth preparedness and complication readiness. However, the BPCR among pregnant women in Bahi, was generally very low.

This implies that the increase in the practice of BPCR was attributed by the intervention. Therefore, community engagement using M-MAMA Champions is effective in improving the practice of birth preparedness and complication readiness among pregnant women as supported by other studies [22]. The intervention utilized women groups to enhance the BPCR practice. The addition of the sensitisation package provided to pregnant women might have also influenced the increase in practice, besides the participatory learning and action intervention model whereby the M-MAMA Champions played their role as facilitators. Therefore, this signifies that, for an effective intervention delivery through the PLA model, the addition of a reference material enhances creating awareness and practice to pregnant women on BPCR.

## Strengths and limitations of the study

The study involved the local authorities through Village Executive Officers and community health workers during the introduction of the study, the intervention was accepted and delivered to the intervention arm as a necessity to save women's lives. M-MAMA Champions were selected from within the localities, which made the intervention acceptable, easier and friendly to deliver and well applauded by the healthcare workers in the intervention clusters. Also, the interaction between pregnant women and the M-MAMA Champions was stronger within the intervention clusters.

The use of M-MAMA Champions as facilitators, based on their experience increased the trust of pregnant women to the intervention for it was anticipated to improve their health and save their lives. The use of a friendly and well-understood package might have increased their enthusiasm to participate in the sensitisation sessions and engagement of their partners while at home to improve their awareness.

Nevertheless, the intervention modality of this research was designed to be delivered every two weeks for a total of three sessions. The two-week interval for the intervention delivery was deferred to four weeks following the harvest season that was ongoing during the study period. The four-week interval of the intervention delivery was more appropriate to increase the study participants' compliance (with as minimal attrition rate as possible). The decision might have compromised the intervention's far better performance. Harvest activities might have contributed to the 28% and 20% attrition rates in the control and intervention arms respectively. A limited duration of only one month for intervention delivery to evaluation doesn't guarantee a long-lasting sustained change in awareness and practices.

The intervention delivery location was designed to be within the community, however due to sparsely located residential areas and a few eligible M-MAMA Champions to reach the distant residing pregnant women, the intervention was delivered at a nearby health facility which was more convenient for the pregnant women to meet for the sensitisation instead of one of the pregnant woman's house. This modality might have influenced the healthcare workers to intensify the standard care delivery including education sessions that might have been infrequently delivered and contributed to the observed outcome.

The M-MAMA Champions with a minimum primary education, might have faced a challenge in understanding the package that was delivered for one day, and the orientation duration might also have contributed to the inadequate understanding of the package among M-MAMA Champions. Inadequate understanding of the package among M-MAMA Champions with low education levels might have impaired the intervention performance.

The intervention package adapted from the Ministry of Health, was translated to Swahili for easy understanding among pregnant women. However, the translation involved subject matter experts without language translators. This might have jeopardized the retention of the true meanings of the phrases or words in the translated package. It might have contributed to the poor understanding of the package among M-MAMA Champions and pregnant women. The intervention was delivered for one month, with only two sessions held, this might not be enough to ensure pregnant women fully understand the ODS, and BPCR for effective practice referring to the minimum of 8 ANC which reflects eight educational program sessions as recommended by WHO.

## Conclusion

Awareness of ODS, and BPCR among pregnant women is the basis for practicing BPCR that ultimately increases the utilization of skilled birth services. Women groups, as has been evidenced in other studies, are a useful model for community engagement to improve awareness

of obstetric danger signs, birth preparedness and complication readiness among pregnant women. Coupled with the women groups in which M-MAMA Champions acted as facilitators, is the use of supporting and friendly material or package contextualized to the local needs (language and pictures) that enables the M-MAMA Champions in the women groups to better sensitize the target audience. M-MAMA Champions, the women groups' facilitators are very effective in enhancing awareness and practice of birth preparedness and complication readiness among pregnant women, therefore, its utilization is deemed to be effective. This implies that community engagement using M-MAMA Champions can easily and effectively be adopted by various stakeholders in the implementation of health programs targeting health education or promotion in the country. However further evaluation towards the end of pregnancy would have accurately determined the effectiveness of the intervention at delivery, it is critical to be replicated in future similar studies. Also, pregnant women's experiences encountered during pregnancy that impair their birth preparedness can be further explored through qualitative approaches.

Patient and Public Involvement; The public was involved in the process of project implementation, whereby the CHWs were part of the project observation. Also, the M-MAMA Champions, from the intervention clusters were involved in implementing the tested intervention.

## Operational definitions

*M-MAMA referral and emergency system* refer to the systems that transport pregnant women from either community or lower to higher-level facilities using ambulances or privately-owned vehicles (community drivers) for definitive care, to reduce second delay, an attribute to maternal death.

*M-MAMA Champions* refers to the mothers who have received the M-MAMA referral and emergency services during their previous delivery, who are ready and willing to empower other pregnant women, residing their communities.

*Community engagement using M-MAMA Champions:* refers to the utilization of women who are the beneficiaries of the M-MAMA program to mobilize pregnant women in their communities to create awareness and improve their knowledge on obstetric danger signs, birth preparedness and complication readiness in the form of women groups, whereby the M-MAMA Champions act as facilitators.

## Acknowledgements

The research authors would like to thank PATHFINDER for sharing crucial information on the M-MAMA referral and emergency system in Tanzania and the Department of Public Health of the University of Dodoma for the guidance in the implementation and conducting the study.

## Author contributions

**Conceptualization:** Alex Sanga, Stephen Kibusi, James Tumaini Kengia.

**Data curation:** Alex Sanga, Stephen Kibusi, James Tumaini Kengia.

**Formal analysis:** Alex Sanga, James Tumaini Kengia.

**Funding acquisition:** Alex Sanga.

**Investigation:** Alex Sanga, James Tumaini Kengia.

**Methodology:** Alex Sanga, Stephen Kibusi, James Tumaini Kengia.

**Project administration:** Alex Sanga, James Tumaini Kengia.

**Resources:** Stephen Kibusi.

**Supervision:** Alex Sanga, Stephen Kibusi, James Tumaini Kengia.

**Validation:** Alex Sanga, Stephen Kibusi, James Tumaini Kengia.

**Visualization:** Alex Sanga, Stephen Kibusi, James Tumaini Kengia.

**Writing – original draft:** Alex Sanga.

**Writing – review & editing:** Alex Sanga, Stephen Kibusi, James Tumaini Kengia.

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
