## [Decision Letter · Decision Letter 0]

13 Nov 2024

PGPH-D-24-02328

Effectiveness of community engagement using M-Mama Champions in improving Awareness of Obstetric Danger Signs, Birth Preparedness and Complication Readiness Among Pregnant Women in Bahi, Dodoma. Cluster Randomized Pragmatic Implementation Trial.

Dear Dr. Sanga,

Thank you for submitting your manuscript to PLOS Global Public Health. After careful consideration, we feel that it has merit but does not fully meet PLOS Global Public Health’s publication criteria as it currently stands. Therefore, we invite you to submit a revised version of the manuscript that addresses the points raised during the review process.

Kindly address the reviewer comments 

We look forward to receiving your revised manuscript.

Kind regards,

Ejemai Eboreime, MD, MSc, PhD

Academic Editor

Journal Requirements:

1. Please ensure that the Title in your manuscript file and the Title provided in your online submission form are the same.

Additional Editor Comments (if provided):

Reviewers' comments:

Reviewer's Responses to Questions

**Comments to the Author**

1. Does this manuscript meet PLOS Global Public Health’s publication criteria ? Is the manuscript technically sound, and do the data support the conclusions? The manuscript must describe methodologically and ethically rigorous research with conclusions that are appropriately drawn based on the data presented.

Reviewer #1: Yes

Reviewer #2: Partly

2. Has the statistical analysis been performed appropriately and rigorously?

Reviewer #1: Yes

Reviewer #2: I don't know

3. Have the authors made all data underlying the findings in their manuscript fully available (please refer to the Data Availability Statement at the start of the manuscript PDF file)?

Reviewer #1: Yes

Reviewer #2: Yes

4. Is the manuscript presented in an intelligible fashion and written in standard English?

Reviewer #1: Yes

Reviewer #2: Yes

5. Review Comments to the Author

Reviewer #1: Its a very well written and expansive manuscript. The research topic is very relevant especially to developing countries. For recruitment of participants, the parity of pregnant women was not considered, it would be nice to explain if bias resulting from education or awareness from previous pregnancies was taken into consideration and if participants were asked if they received BPCR education in previous pregnancies?

Also, there are lot of acronyms used in the manuscript, so it would good to have the full form either mentioned in a list or in the manuscript.

Reviewer #2: Thanks for requesting that I review this paper.

The study investigates the effectiveness of community engagement through M-Mama Champions in raising awareness of obstetric danger signs and improving birth preparedness and complication readiness among pregnant women in Dodoma, Tanzania.

The research is relevant and aligns with global maternal and child health priorities, especially in rural areas where maternal health education is critical to reducing maternal and child mortality.

Areas requiring reviews that could strengthen the study

Choice of facility

The study is conducted in rural wards of Bahi, Dodoma, Tanzania. Facility selection includes four wards, each with at least two healthcare facilities (either health centers or dispensaries). The choice of these rural facilities is well justified given the focus on reaching underserved populations where the risk of maternal mortality is higher. The inclusion of both health centers and dispensaries appears to address variability in resources and access within the community, providing a representative look at maternal healthcare challenges in rural settings.

Clarify Facility Criteria: The rationale for selecting specific wards with particular facility types could be expanded. For instance, including how facility resources, personnel, or services influenced their selection would highlight how these differences might impact intervention outcomes.

Explanation of Facility Distribution: Since two wards were randomly assigned to the intervention or control arms, an explanation of how the assignment affected facility distribution (if relevant) would provide additional clarity. If geographic proximity to other healthcare services or resource availability differed among wards, this could have implications for intervention efficacy.

Study Participants

The study includes pregnant women in their first and second trimesters (gestational age up to 28 weeks). Selection was randomized, with an equal number of participants from each facility

Inclusion and Exclusion Criteria: The inclusion criteria are clearly defined (first and second trimesters), and the study excludes women with serious health conditions. While the criteria help focus on early pregnancy education, expanding the rationale for excluding women in later trimesters or addressing how this might affect birth preparedness insights would be helpful.

Remoteness and Community Awareness: Considering the members of these communities, despite all rural settings, the level of remoteness of each site also plays a significant role in health and social awareness. Generally, less remote areas tend to have greater access to information and services, leading to a higher level of health and social awareness compared to more isolated locations.

Review Bias in Outcome assessment : Losses to follow-up due to adverse pregnancy events could bias the study findings by excluding cases that might reveal gaps in the intervention's effectiveness or identify additional needs that were not met. This could lead to an overly optimistic assessment of the intervention’s success if only healthier or less complex cases remain in the study.

I would recommend including a mechanism to track reasons for loss to follow-up as this could provide critical insights into whether adverse outcomes are contributing to study dropout. If feasible, a follow-up to gather information on lost cases would strengthen the study’s conclusions and provide a more comprehensive picture of maternal health needs in the community.

Limited Intervention Duration and Follow-up: The intervention’s duration (one month) may not provide sufficient time to instill lasting changes in awareness and practices. A short follow-up period might capture only the immediate effects of the intervention, without insight into how well these practices are retained over time or during delivery. Given the importance of sustained knowledge and readiness, a follow-up at later stages (e.g., late pregnancy or postpartum) could provide stronger evidence of long-term efficacy and any adjustments needed for sustained impact.

Confounding Variables : Though this study was in a rural context- confounding variables, such as socioeconomic status, access to transportation, or cultural beliefs about childbirth, which could impact awareness and preparedness levels were not reviewed for or assessed . it’s difficult to determine how much of the improvement in awareness and preparedness is due to the intervention versus other factors.

6. PLOS authors have the option to publish the peer review history of their article (what does this mean? ). If published, this will include your full peer review and any attached files.

**Do you want your identity to be public for this peer review?** For information about this choice, including consent withdrawal, please see our Privacy Policy .

Reviewer #1: **Yes: ** Meenakshi Bhilwar

Reviewer #2: No

---

## [Decision Letter · Decision Letter 1]

30 Jan 2025

PGPH-D-24-02328R1

The effectiveness of community engagement using M-Mama Champions in improving Awareness of Obstetric Danger Signs, Birth Preparedness and Complication Readiness Among Pregnant Women in Bahi, Dodoma.

Dear Dr. Sanga,

Thank you for submitting your manuscript to PLOS Global Public Health. After careful consideration, we feel that it has merit but does not fully meet PLOS Global Public Health’s publication criteria as it currently stands. Therefore, we invite you to submit a revised version of the manuscript that addresses the points raised during the review process.

The reviewer is not satisfied with your response to their previous concerns. Kindly look again at the comments and provide a rebuttal where necessary.

We look forward to receiving your revised manuscript.

Kind regards,

Ejemai Eboreime, MD, MSc, PhD

Academic Editor

Journal Requirements:

Additional Editor Comments (if provided):

The reviewer is not satisfied with your response to their previous concerns. Kindly look again at the comments and revise or provide a rebuttal where necessary.

Reviewers' comments:

Reviewer's Responses to Questions

**Comments to the Author**

1. If the authors have adequately addressed your comments raised in a previous round of review and you feel that this manuscript is now acceptable for publication, you may indicate that here to bypass the “Comments to the Author” section, enter your conflict of interest statement in the “Confidential to Editor” section, and submit your "Accept" recommendation.

Reviewer #1: (No Response)

2. Does this manuscript meet PLOS Global Public Health’s publication criteria ? Is the manuscript technically sound, and do the data support the conclusions? The manuscript must describe methodologically and ethically rigorous research with conclusions that are appropriately drawn based on the data presented.

Reviewer #1: Yes

3. Has the statistical analysis been performed appropriately and rigorously?

Reviewer #1: Yes

4. Have the authors made all data underlying the findings in their manuscript fully available (please refer to the Data Availability Statement at the start of the manuscript PDF file)?

Reviewer #1: Yes

5. Is the manuscript presented in an intelligible fashion and written in standard English?

Reviewer #1: Yes

6. Review Comments to the Author

Reviewer #1: (No Response)

7. PLOS authors have the option to publish the peer review history of their article (what does this mean? ). If published, this will include your full peer review and any attached files.

**Do you want your identity to be public for this peer review?** For information about this choice, including consent withdrawal, please see our Privacy Policy .

Reviewer #1: No

---

## [Editor Report · Decision Letter 2]

6 Feb 2025

The effectiveness of community engagement using M-Mama Champions in improving Awareness of Obstetric Danger Signs, Birth Preparedness and Complication Readiness Among Pregnant Women in Bahi, Dodoma.

PGPH-D-24-02328R2

Dear Dr. Sanga,

We are pleased to inform you that your manuscript 'The effectiveness of community engagement using M-Mama Champions in improving Awareness of Obstetric Danger Signs, Birth Preparedness and Complication Readiness Among Pregnant Women in Bahi, Dodoma.' has been provisionally accepted for publication in PLOS Global Public Health.

Best regards,

Ejemai Eboreime, MD, MSc, PhD

Academic Editor